# Mitigating Object Hallucination in Large Vision Language Model with Human-Free Reinforcement Learning

## Abstract

Large Vision-Language Models (LVLMs) have excelled in joint visual and language understanding, particularly in generating detailed image captions. However, they still struggle with object hallucination, where non-existent objects are described, especially in long captions. While fine-tuning through supervised learning with enhanced datasets or reinforcement learning from human feedback can alleviate this issue, these methods demand considerable human effort, limiting scalability. This paper addresses this challenge by introducing a human-free framework to mitigate object hallucination in LVLMs for image captioning, utilizing reinforcement learning driven exclusively by automatic natural language processing metrics. We demonstrate that the following framework can effectively mitigate hallucination: (1) caption generation is formulated as a Markov Decision Process (MDP); (2) minimizing hallucination while maintaining caption quality is guided by a reward function, combining a proposed *F1Score* with a penalty on Kullback–Leibler divergence from the pre-trained model; (3) fine-tuning the LVLM within the MDP framework can be performed directly by Proximal Policy Optimization (PPO) with careful attention to architectural details. Extensive experiments demonstrate a significant reduction in hallucination by up to 41% while preserving the caption quality compared to the baseline model, InstructBLIP, on the COCO dataset. This improvement is reflected in consistent gains in object coverage and accuracy across various models and datasets. Notably, our method achieves comparable or superior performance to alternative approaches, all without requiring any human involvement.

## 1 Introduction

Large Vision-Language Models (LVLMs) have become increasingly prominent due to their ability to perform joint visual and language understanding tasks Achiam et al. (2023); Alayrac et al. (2022). Among these, image captioning has emerged as a key application where LVLMs consistently outperform smaller models by generating highly detailed and contextually rich captions Dai et al. (2023); Zhu et al. (2023); Li et al. (2023a). Despite these advancements, LVLMs still struggle with a critical challenge: object hallucination Rohrbach et al. (2018b); Biten et al. (2022). This occurs when captions include references to objects that do not exist in the corresponding image, particularly in longer, more detailed descriptions; as shown in Fig. 1. Object hallucination not only undermines the credibility of these models but also hinders their broader application in fields that require high precision, such as autonomous systems and medical imaging.

Addressing object hallucination has been a major focus in recent research efforts Zhou et al. (2023); Li et al. (2023d); Dai et al. (2022); Liu et al. (2024). Early efforts aimed at mitigating this issue in small-scale multimodal pre-trained models focused on reducing object co-occurrence patterns through data augmentation Biten et al. (2022); Rohrbach et al. (2018b); Kim et al. (2023). However, such approaches were considered ineffective for LVLMs Zhou et al. (2023). More recent studies have explored improving dataset quality and applying fine-tuning to LVLMs Gunjal et al. (2023); Li et al. (2023c); Liu et al. (2023a), or using Reinforcement Learning from Human Feedback (RLHF) Sun et al. (2023) to reduce object hallucination. Despite their potential, these methods still face significant challenges, as gathering large volumes of high-quality examples Gunjal et al. (2024); You

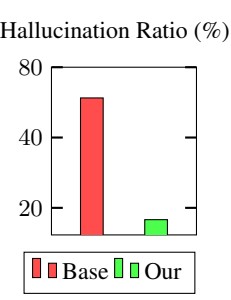 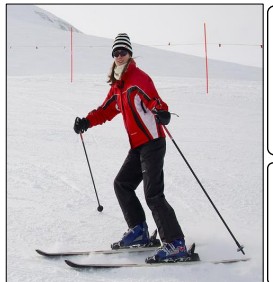

Hallucination Ratio (%) ↓

**Base:** A skier in a bright red jacket and black pants swiftly descends a snowy slope, expertly navigating through a series of orange flags. In the background, a **snowboarder** performs a trick off a ramp, while a **group of spectators** cheer from the sidelines. **Snow-covered trees** line the slope, and a **chairlift** carries skiers up the mountain.

**Our:** A person skiing on a snow-covered slope, wearing a red jacket, black pants, and using poles to navigate. The skier is alone on the slope, with no other people visible in the scene. The background is filled with snow creating a serene and focused atmosphere.

(a) Sentence hallucination ratio measured in $CHAIR_s$ (b) A detailed caption example from COCO dataset for baseline (Base) vs fine-tuned (Our). Bold objects are hallucinated ones by LVLMs.

Figure 1: Quantitative and qualitative comparison between Base (InstructBLIP) and Our: Chart (a) shows a significant 41% reduction in object hallucination on the COCO dataset using Our. Fig. (b) presents an example where the Base model produces a caption with substantial object hallucination, while the Our model provides an accurate description without hallucinated objects.

et al. (2023); Zhang et al. (2024) or obtaining accurate human feedback for RLHF fine-tuning Stiennon et al. (2020) remains a time-consuming and labor-intensive process that requires considerable human expertise and effort.

To address these limitations, we propose a human-free framework to mitigate object hallucination in LVLMs for image captioning. Our approach leverages reinforcement learning, guided exclusively by automatic natural language processing (NLP) metrics, eliminating the need for human intervention. The key features of our framework are as follows:

- **Caption Generation as an MDP**: To streamline previous methods and minimize human intervention, we formulate the caption generation task as a Markov Decision Process (MDP), with a reward function incorporating specific automatic NLP metrics to reduce hallucination. By framing image captioning as a reinforcement learning problem, we can effectively address the inherent non-differentiability challenge of optimizing automatic metrics, which are difficult to optimize directly through traditional supervised learning methods.

- **Dedicated Reward Function**: To guide the output generation behavior, we incorporate automatic NLP metrics into the reward function. For hallucination reduction, instead of using the straightforward *CHAIR* metric Rohrbach et al. (2018b), we introduce *F1Score*, which provides a better balance between reducing object hallucination and improving object coverage. Additionally, we introduce a Kullback–Leibler (KL) divergence penalty to prevent the policy from diverging too far from the pre-trained model, preserving caption quality without the need for labeled data. Moreover, since metrics like *F1Score* are computed only at the end of caption generation, which results in sparse rewards, the KL penalty helps densify feedback, making RL optimization more effective. Optionally, when labeled data is available, the reward can easily adopt other quality metrics such as *Meteor* Banerjee & Lavie (2005) and *BERTScore* Zhang et al. (2019) to further improve caption quality.

- **Efficient Fine-tuning with PPO:** The proposed framework can be directly optimized using Proximal Policy Optimization (PPO), a popular RL method, to fine-tune the Large Vision-Language Model (LVLM). However, training LVLMs typically requires significant memory and computational resources. To mitigate this, we introduce a compact version of PPO where the policy, value function, and reference model share the same frozen language model, adding only minimal additional training parameters through adapters. These adapters are compatible with recent state-of-the-art fine-tuning techniques for LVLMs such as prompt tuning, ensuring resource-efficient training.

Through extensive experiments, we demonstrate that our method reduces hallucination by up to 41% compared to the baseline model, InstructBLIP, while also improving object coverage and caption quality on the COCO dataset. Additionally, our framework can be easily extended to handle more complex datasets (e.g. Visual Genome) and incorporate existing NLP metrics effectively. Notably,

our approach achieves comparable or superior performance to existing methods, all without relying on human feedback, making it a scalable and efficient solution for enhancing LVLMs in image captioning tasks.

## 2 RELATED WORK

**Large Vision Language Model:** The rapid advancements in Large Language Models (LLMs) Touvron et al. (2023); Chung et al. (2022); Touvron et al. (2023) combined with a surge in open-source initiatives, has paved the way for the emergence of extensive vision-language models Liu et al. (2023c); Zhu et al. (2023); Sun et al. (2023); Ye et al. (2023); Bai et al. (2023); Peng et al. (2023). LVLMs seamlessly combine a LLM and a pre-trained visual encoder to form an end-to-end model, aiming to produce contextually relevant text from visual stimuli Zhang et al. (2023a). There are various approaches to effectively achieve this. LLaVA Liu et al. (2023b) introduced the concept of integrating a simple projector during LLM fine-tuning. Chatspot Zhao et al. (2023) follow LLaVA's model structure, embeds the region of interest into instruction data. GPT4RoI Yu et al. (2023) and Shikra Chen et al. (2023) add grounding tasks to LLaVA structure models and achieve great performance on various tasks. Concurrently, Multimodal-GPT Gong et al. (2023) aims to improve OpenFlamingo's Alayrac et al. (2022) directive adherence. mPLUG-Owl Ye et al. (2023) suggests a two-step method: first train vision models, and then refine the language model using techniques like LoRA Hu et al. (2021). BLIP2 Li et al. (2023b) and InstructBLIP Dai et al. (2023) presented Q-former-based LVLMs without fine-tuning the LLM but achieving state-of-the-art performance. Our work fine-tunes the InstructBLIP to reduce object hallucination within LVLMs.

**Object Hallucination in Vision Language Models:** Object hallucination refers to generated descriptions containing objects which are not present in the visual modality Rohrbach et al. (2018b). In small-scale vision language models (VLM), mitigation techniques include fine-grained contrastive learning Zeng et al. (2021) or data augmentation to eliminate co-occurrence patterns Kim et al. (2023). However, training paradigms differ between conventional VLMs and LVLMs. The autoregressive training paradigm in LVLMs poses challenges in implementing VLM hallucination mitigation methods directly Zhang et al. (2023b). Notably, object hallucination is more pronounced and widespread in the long-form descriptions produced by LVLMs compared to the shorter descriptions generated by VLMs. Ongoing research has started to tackle object hallucination in LVLMs, encompassing evaluation and detection approaches Petryk et al. (2024); Li et al. (2023d); Liu et al. (2023a); Dai et al. (2022); Jing et al. (2023); Liu et al. (2023a); Sun et al. (2023), the development of benchmarks Ben-Kish et al. (2024); Wang et al. (2023), hallucination elimination through the construction of higher-quality datasets Gunjal et al. (2023); Li et al. (2023c); You et al. (2023), and the use of supervised learning for fine-tuning Zhou et al. (2023); Zhai et al. (2023) or employ Reinforcement Learning training from Human Feedback (RLHF) Sun et al. (2023); Stiennon et al. (2020) to align different modalities. However, these methods often demand substantial time and labor, particularly in acquiring a large number of high-quality examples. Instead, grounded in reinforcement learning (RL) and automatic metrics, we propose a novel approach. This conceptually distinct method demonstrates efficacy in reducing hallucination and is compatible with various LVLMs, offering a more efficient solution without relying on human effort.

**Reinforcement Learning for NLP:** Reinforcement Learning (RL) has emerged as a prevalent technique for enhancing language models in a wide range of Natural Language Processing (NLP) tasks, encompassing dialogue Li et al. (2016); Zhou et al. (2017); Jaques et al. (2019); Yi et al. (2019); Jaques et al. (2020), machine translation Wu et al. (2016); Nguyen et al. (2017); Kiegeland & Kreutzer (2021); Bahdanau et al. (2016); Ranzato et al. (2015); Kreutzer et al. (2018), image captioning Rennie et al. (2017); Ren et al. (2017), summarization Stiennon et al. (2020); Paulus et al. (2017); Wu & Hu (2018); Bohm et al. (2019); Ziegler et al. (2019), and text-games Narasimhan et al. (2015); Hausknecht et al. (2020). In this training paradigm, NLP models are optimized through an RL algorithm, wherein the reward signal is derived from either human feedback Kreutzer et al. (2018); Jaques et al. (2020); Stiennon et al. (2020); Ziegler et al. (2019) or NLP evaluation metrics, such as ROUGE for summarization Paulus et al. (2017); Wu & Hu (2018) or BLUE for translation Wu et al. (2016); Nguyen et al. (2017); Kiegeland & Kreutzer (2021). These reward mechanisms enable the models to iteratively improve and fine-tune their performance based on the quality of generated outputs. While RL has proven effective in NLP, its exploration in Vision Large Language Models (LVLMs) for captioning is not well-established. Our work pushes the boundaries in this

direction by leveraging RL to address the challenge of object hallucination in LVLMs. We tackle intricate issues specific to this context, including high computational costs, sparse rewards, and extended temporal horizons.

**Finetuning LVLMs with Adapters:** Fine-tuning the entire model for Large Vision Language Mode demands extensive memory and computational resources. To address this challenge, various Parameter Efficient Fine-Tuning (PEFT) methods have emerged as cost-effective alternatives. These methods include prompt tuning Lester et al. (2021); Li & Liang (2021); Qin & Eisner (2021), tuning the embedding layer inputs An et al. (2022), tuning hidden states (IA3 ) Liu et al. (2022), employing Low-rank Adapters (LoRA) Hu et al. (2021); Dettmers et al. (2023), incorporating full layers Houlsby et al. (2019), tuning biases Zaken et al. (2021), learning weight masks based on Fisher information Sung et al. (2021), and leveraging combinations of these approaches Karimi Mahabadi et al. (2021). In our study, we demonstrate the effectiveness of prompt tuning in addressing the task at hand, while future work will investigate trade-offs with other PEFT methods to further enhance performance.

# 3 METHODOLOGY

In this session, we will sequentially cover the following topics: (1) casting the caption generation task within the framework of a Markov Decision Process (MDP); (2) defining the dedicated reward function with appropriate automatic metrics; (3) modeling RL networks; (4) fine-tuning the model by solving the MDP through Proximal Policy Optimization (PPO).

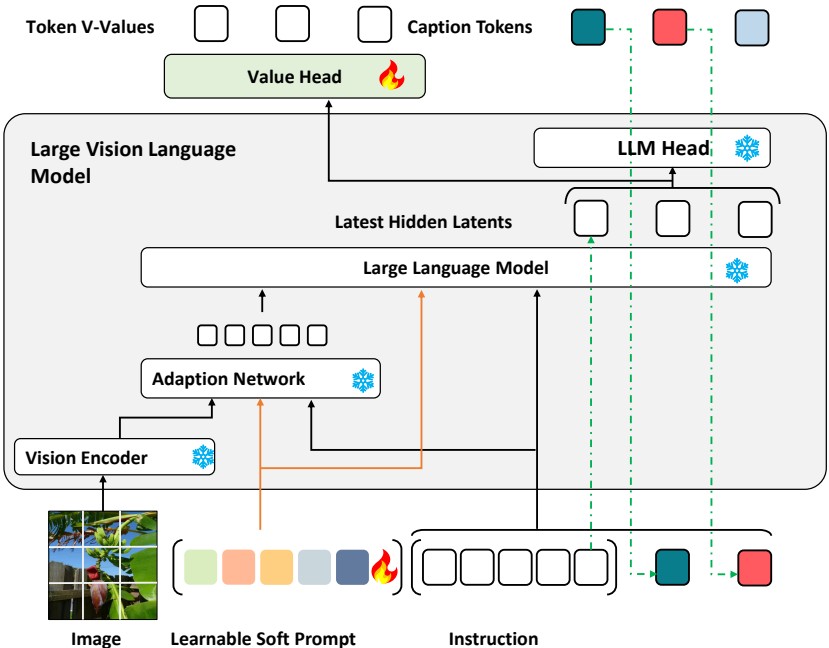

Figure 2: Detailed architecture of our framework. Specifically, the Policy Network is crafted by augmenting the shared LVLM with delicately learnable soft prompts. Meanwhile, the Value Network is formed by replacing the LLM Head with a Linear Value head. Notably, all parameters of the LVLM remain shared and frozen, with only a very small fraction (less than 0.01% LVLM weight) of trainable parameters added to the LVLM for the meticulous modeling of the policy network and value network.

## 3.1 MARKOV DECISION PROCESS (MDP) FOR IMAGE CAPTIONING

The image captioning task can be effectively framed as an MDP due to its inherent sequential nature, where each token generation is a decision based on the current state. This allows us to utilize RL techniques to optimize caption quality holistically, addressing both local and global aspects

of the generated text. Mathematically, we formulate the image captioning as an MDP denoted by $\langle \mathcal{S}, \mathcal{A}, \mathcal{R}, P, \gamma, H \rangle$. Each episode in this MDP begins by sampling a datapoint $(X, Z, Y)$ from our dataset $\mathcal{D} = \{(X_i, Z_i, Y_i)\}_{i=1}^N$, where $X \in \mathcal{X}$ represents the text input for LVLMs, $Z \in \mathcal{Z}$ represents the image, and $Y \in \mathcal{Y}$ is the ground truth caption, which can be set to *none* if no ground truth caption is available. The initial state $S_0 = (Z, x_0, \cdots, x_m)$ consists the image $Z$ and the text input $X = (x_0, \cdots, x_m)$, where $S_0 \in \mathcal{S}$ and the state space $\mathcal{S} = \mathcal{Z} \cup \mathcal{X}$ is defined as the concatenation of images and text inputs. At each time step $t$, an action $a_t \in \mathcal{A}$, which corresponds to a token from our vocabulary $\mathcal{V}$, is taken in the environment from a policy (e.g. an LVLM). The transition function $P : \mathcal{S} \times \mathcal{A} \to \Delta(\mathcal{S})$ deterministically appends an action $a_t$ to the end of the state $S_{t-1} = (Z, x_0, \cdots, x_m, a_0, \cdots, a_{t-1})$ to form the state $S_t$. This process continues until the end of the episode $t \leq T \leq H$, either when the current time step $t$ exceeds the horizon $H$ or when an end-of-sentence (EOS) token is generated, resulting in a final state $S_T = (Z, x_0, \cdots, x_m, a_0, \cdots, a_T)$. At every step, a reward $\mathcal{R} : \mathcal{S} \times \mathcal{A} \times \mathcal{Y} \to \mathbb{R}^1$ is emitted. This reward may be derived from automated metrics (e.g., CHAIR). Our objective is to maximize the cumulative return represented by the equation:

$$\max_{A=\{a_0 \ldots a_T\} \in \mathcal{V}^T} \sum_t \gamma^t \mathcal{R}\left(S_t, a_t, Y\right). \tag{1}$$

where $\gamma$ denotes the discount factor (e.g., 0.99) and $A$ is the generated caption from the LVLM.

## 3.2 Reward function

To tackle hallucination, the first approach people usually think of is to incorporate *CHAIR_i* and *CHAIR_s* Rohrbach et al. (2018a) directly into the reward function. Although *CHAIR* metrics primarily evaluate precision, they cause models to prioritize precision at the expense of recall. To address this issue, we propose utilizing the *F1Score*. *F1Score* offers a balanced measure of precision and recall, ensuring that the reward function encourages comprehensive object coverage while maintaining accuracy:

$$F1Score = \frac{2 * Precision * Recall}{Precision + Recall} \tag{2}$$

where $Precision$ is the ratio of correct objects to all predicted objects, and $Recall$ is the ratio of correct objects to all objects in the ground truth. The ground truth objects can be either extracted using an off-the-shelf object detection model (e.g., YOLOv8 Varghese & Sambath (2024)) or obtained directly from the dataset. Predicted objects can be easily extracted from the caption using a method similar to *CHAIR* Rohrbach et al. (2018a).

The resulting reward function is:

$$\mathcal{R}\left(S^t, a^t, Y\right) = \begin{cases} F1Score(S^T, a^T, Y) \text{ if } t = T \\ 0 \text{ otherwise.} \end{cases} \tag{3}$$

Optionally, in the setting where ground truth captions are available, two additional metrics can be integrated into the reward function to further enhance caption quality: *Meteor* Banerjee & Lavie (2005) and *BERTScore* Zhang et al. (2019). *Meteor* evaluates the similarity between generated and reference texts (a.k.a ground truth captions) based on n-grams and word order, ensuring structural and lexical alignment. Meanwhile, *BERTScore* assesses semantic similarity using pre-trained BERT embeddings, capturing underlying meaning accurately. Together, *Meteor* and *BERTScore* offer a comprehensive evaluation of caption quality, considering both surface-level and semantic aspects, thereby improving caption relevance to the ground truth.

The enhanced reward function is defined as:

$$\mathcal{R}\left(S^t, a^t, Y\right) = \tag{4}$$

$$\begin{cases} F1Score(S^T, a^T, Y) + \alpha Meteor(S^T, a^T, Y) + \beta BERTScore(S^T, a^T, Y) \text{ if } t = T \\ 0 \text{ otherwise.} \end{cases} \tag{5}$$

Note that the balancing weight $\alpha$ for *Meteor* should be relatively smaller compared to *F1Score* and *BERTScore*, as it may encourage shorter captions, especially in datasets with shortened ground truth references.

## 3.3 Modeling Reinforcement Learning Networks

To fine-tune the LVLM within our MDP framework, we utilize a policy network, a value network, and a reference network. While the policy and value networks are essential components of RL, the reference network serves as a proposed teacher network. Its role is to prevent the policy network from deviating too far from the baseline during training, which is particularly important for preserving the caption meaning in the absence of ground truth captions. Given the intensive computational demands of conventional LVLM fine-tuning, we design lightweight and efficient networks. Fig. 3 displays the simplified overview of the framework's network components. Specifically, each network builds upon the same frozen LVLM

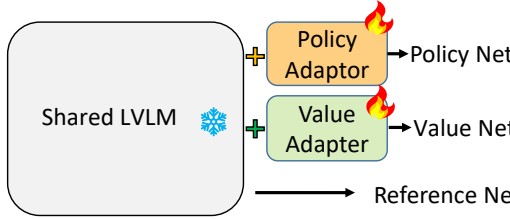

Figure 3: Simplified overview of the framework's network components: policy network, value network, and reference network. All networks share the same frozen LVLM as its foundation. The reference network mirrors the LVLM identically, while the value and policy networks incorporate a lightweight adapter into the shared LVLM.

foundation. The reference network mirrors the LVLM identically, while the value and policy networks incorporate slender adapters alongside the LVLM. This approach optimizes computational resources and is compatible with various state-of-the-art Parameter Efficient Fine-Tuning (PEFT) methods Mangrulkar et al. (2022), which rely on adapters.

In this paper, we utilize *Prompt Tuning* to assess the framework's effectiveness. Prompt Tuning offers an efficient and flexible method for controlling LVLM behavior. By allowing the model to remain frozen while refining prompts, this approach reduces computational costs and provides task-specific adaptability without compromising the model's generalization capabilities. Specifically, the LVLM generates captions based on images and instructions in an autoregressive manner. By prefixing a controllable prompt to the instruction, we can influence the model's behavior Lester et al. (2021). Mathematically, we adopt a conditional generation perspective, where A represents a sequence of tokens forming a caption. The captioning process by LVLM is expressed as $P_\theta(A|X, Z)$, with $\theta$ denoting the LVLM's weight. Prompting enhances the model's generation of A by providing additional context, which is achieved by prefixing a token sequence G to the input X. This aids the model in improving the likelihood of generating the ground truth caption Y: $P_\theta(Y|[G; X], Z)$. Throughout, the model parameters $\theta$ remain unchanged. Optimal G selection can be achieved via manual exploration (*Hard Prompting*) or by representing G with dedicated parameters $\phi$, refined through gradient descent (*Soft Prompting*). This updates the conditional generation as $P_{\theta;\phi}(A|[G; X], Z)$, trainable by maximizing reward through backpropagation, with gradient updates solely applied to $\phi$, i.e., learnable soft prompt.

Fig. 2 illustrates the detailed architecture of the Augmented LVLM in our implementation. The Policy Network $\pi_{\theta;\phi}(A|G, S)$, identical to $P_{\theta;\phi}(A|[G; X], Z)$, is constructed by enhancing the shared large language model with the delicately *learnable soft prompt*. Concurrently, the Value Network $V_{\theta;\omega}(S)$ is created by substituting the LLM Head with a Value Head, featuring a single output neuron. The reference network $\pi_\theta^r(A|S)$ remains identical to the original LVLM. Notably, all parameters of the Large Vision Language Model persist as shared and frozen. Only an extremely small fraction (approximately 0.01% LVLM weight) of trainable parameters is introduced to meticulously model the policy network and value network.

### 3.4 FINE-TUNING MODEL BY SOLVING THE MDP

Given the MDP and the RL networks, we fine-tune the augmented LVLM, i.e., the policy, using the on-policy Proximal Policy Optimization (PPO) algorithm Schulman et al. (2017). Formally, this algorithm trains the policy $\pi_{\theta;\phi}(A|G, S)$ to maximize long-term discounted rewards over generated captions:

$$\mathbb{E}_\pi \left[ \sum_{t=0}^{T} \gamma^t \mathcal{R}\left(S_t, a_t, Y\right) \right]. \tag{6}$$

We define our V-value and Q-value functions as follows:

$$V^\pi(S_t) = \mathbb{E}_{a_t \sim \pi, Y \sim \mathcal{D}} \left[ \sum_{\tau=t}^{T} \gamma^\tau R\left(S_\tau, a_\tau, Y\right) \right] \tag{7}$$

$$Q^{\pi}(S_t, a_t) = \mathbb{E}_{Y \sim \mathcal{D}} R(S_t, a_t, Y) + \gamma \mathbb{E}_{s_{t+1} \sim P}\left[V^{\pi}(S_{t+1})\right]. \tag{8}$$

This leads to the definition of our advantage function:

$$A^{\pi}(S_t, a_t) = Q^{\pi}(S_t, a_t) - V^{\pi}(S_t). \tag{9}$$

We use the previously mentioned value network $V_{\theta;\omega}$ to model the value function, and the mentioned Reference Network $\pi_\theta^r(A|S)$ to generate the initial caption. Following the components defined, we employ the PPO algorithm detailed in Schulman et al. (2017) to fine-tune the policy. To enhance training stability, we approximate the advantage using Generalized Advantage Estimation as outlined in Schulman et al. (2015).

Given a data point tuple $(X, Z, Y)$ and generated caption A from our policy, as the aforementioned environment reward is sequence-level and sparse, we further regularize the reward function using a token-level KL penalty. This penalty ensures the model does not deviate significantly from the original caption generated by $\pi_\theta^r(A|S)$, densifying the reward signal and preserving the quality and meaning of the caption in line with the reference model. This regularization is especially crucial when the ground truth caption $Y$ is unavailable. Formally, the regularized reward function is defined as:

$$\hat{R}(S_t, a_t, Y) = R(S_t, a_t, Y) \tag{10}$$
$$- \lambda \mathrm{KL}\left(\pi_\theta\left(a_t \mid G, S_t\right) \| \pi^r\left(a_t \mid S_t\right)\right). \tag{11}$$

Here, $\hat{R}$ is the regularized KL reward, KL denotes Kullback–Leibler divergence, and the KL coefficient $\lambda$ is dynamically adapted, following Ziegler et al. (2019).

# 4 EXPERIMENT

## 4.1 EXPERIMENTAL SETTINGS

**Datasets:** We train and evaluate our method using the COCO dataset, as described by Lin et al. (2014). This dataset serves as a comprehensive collection widely used in tasks such as image recognition, segmentation, and captioning. It encompasses over 300,000 images, covering more than 80 object categories, and is meticulously annotated. For our captioning task, we utilize the Karpathy split Karpathy & Fei-Fei (2015), dividing the dataset into training, validation, and test sets with 82,000, 5,000, and 5,000 images, respectively. Additionally, to prepare the dataset for LVLM fine-tuning, we randomly augment each image with detailed caption instructions. A complete list of instructions is provided in Appendix G.

**Implementation detail:** We employ InstructBLIP Dai et al. (2023) as our baseline LVLM due to its robust resistance to hallucination compared to others. InstructBLIP adopts the BLIP-2 architecture Li et al. (2023b) and is distinguished by its use of Q-former, a Query Transformer designed for instruction-aware training. In this paper, the vision encoder utilized is ViT-g/14 Fang et al. (2023), while the LLM of choice is Vicuna-7B. During RL fine-tuning, we initialize the model with the pre-trained InstructBLIP checkpoint. Subsequently, we exclusively fine-tune the parameters of our adapters, keeping the image encoder, Q-former, and LLM frozen.

Our experiments are conducted using the Transformers Wolf et al. (2020) and PyTorch Paszke et al. (2019) frameworks. For fine-tuning on the dataset, we employ the same tokenizer as InstructBLIP with vocabulary size $\mathcal{V}$ 32000. Our reward function sets $\alpha$ and $\beta$ to 0.1 and 1, respectively. The soft prompt length is set to 20. In implementing PPO, we adopt the default parameters of the Stable Baseline API Raffin et al. (2021), with modifications: we gather 4096 transitions and update the PPO loss 5 times for each on-policy step. The $\gamma$ is set to 0.99. The KL coefficient $\lambda$ is dynamically adjusted, as described in Ziegler et al. (2019), with a target KL of 0.05. Our batch size is set to 64, and we train the models using the AdamW optimizer with a learning rate of 0.0002, ensuring stable convergence over 50 epochs. We leverage 8 Nvidia A6000 GPUs, employing mixed precision and flash attention mechanisms Dao et al. (2022) to enhance training speed. The fine-tuning process typically requires approximately one day to complete.

## 4.2 EXPERIMENTAL RESULTS

In this section, we present experimental results that highlight five key points: (1) the occurrence of object hallucination and its amplification in detailed captions; (2) the potential of prompt-tuning

(demonstrated using hard prompting) to mitigate hallucination; (3) the effectiveness of our framework in reducing hallucination while preserving or even enhancing caption quality when ground truth captions are available, compared to baseline models and alternative methods; and (4) the framework's robustness when applied to more complex datasets.

**The occurrence of object hallucination and its amplification in detailed captions:** We begin by conducting an experiment aimed at demonstrating the presence of object hallucination and its amplification with detailed captions. We design instructions to generate short and long captions using two baseline models: InstructBLIP and mPLUG-Owl. Tab. 1 illustrates the object hallucination measured by CHAIRs across various caption types with specific input prompts on the COCO test set. The results indicate that LVLMs experience object hallucination for both short and long captions, with the issue being more pronounced for longer captions. Notably, InstructBLIP exhibits less hallucination with short captions; however, the problem amplifies significantly, around ten times, with longer sentences. Both models show similarly high rates of hallucination in long captions demonstrating the severity of the problem.

| Type | Prompt | InstructBLIP | | mPLUG-Owl | |
|------|--------|--------------|--|-----------|--|
| | | $CHAIR_i(\%)\downarrow$ | $CHAIR_s(\%)\downarrow$ | $CHAIR_i(\%)\downarrow$ | $CHAIR_s(\%)\downarrow$ |
| Short | Generate a short caption of the image. | 2.43 | 3.13 | 22.81 | 60.55 |
| | Create a textual summary for the image. | 4.95 | 6.51 | 22.98 | 61.33 |
| Long | Provide a detailed description of the image. | 27.01 | 60.91 | 26.03 | 71.39 |
| | Create a detailed textual summary for the image. | 25.80 | 59.11 | 24.25 | 66.31 |

Table 1: Object Hallucination, gauged by $CHAIR_s$ and $CHAIR_i$ metrics, across diverse caption types paired with specific input prompts in the COCO test set. These prompts are designed to elicit both short and long captions. Two distinct methods are illustrated: InstructBLIP and mPLUG-Owl.

**Prompt tuning in mitigating hallucination:** We have meticulously curated a series of hard prompts intended to be incorporated at the beginning of input instructions, aimed at minimizing object hallucination in the model's generated captions. Each prompt is meticulously designed to address specific sources of object hallucination, strategically guiding the model away from potential pitfalls. The comprehensive list of prompts is provided in Appendix H. During the testing phase, we employ a randomized approach by selecting a single hard prompt to prefix each sample instruction. We conduct captioning using the InstructBLIP baseline model with prefixed instructions. The reported performance metrics reflect the average performance across these instances, focusing particularly on *CHAIR* evaluations as shown in Tab. 4 under the label *Hard Prompting*.

| Method | $CHAIRi(\%)\downarrow$ | $CHAIRs(\%)\downarrow$ |
|--------|------------------------|------------------------|
| mPLUG-Owl | 26.2 | 70.5 |
| LLaVA | 22.5 | 62.7 |
| InstructBLIP | 25.8 | 59.1 |
| Teacher | 7.5 | 36.4 |
| CoT | 7.8 | 35.7 |
| Greedy-Decoding | 7.8 | 35.5 |
| GPT-Ensemble | 13.0 | 51.0 |
| GPT-Teacher | 7.8 | 32.0 |
| Hard Prompting | 20.9 | 45.1 |
| Our | **6.8** | **17.8** |

Figure 4: Performance of Object Hallucination. The first row showcases non-fine-tuned LVLM baselines. The second row features fine-tuning methods referenced in Zhou et al. (2023). The third row illustrates our Hard Prompting on baseline InstructBLIP, while the last row demonstrates our Soft Prompt fine-tuning using our RL framework.

In comparison to InstructBLIP, we observe that hard prompting can mitigate object hallucination by reducing $CHAIR_i$ and $CHAIR_s$ from 25.8 to 20.9 (-4.9%) and 59.1 to 45.1 (-14%) respectively. This highlights the effectiveness of prompt tuning as a method to reduce object hallucination.

**Performance of our framework:** Based on the observed effectiveness of hard prompting, we fine-tuned the InstructBLIP model using a learnable soft prompt within our framework to optimize prompt selection. In Table 4, we present the performance of our proposed method compared to various baselines. The first row represents hallucination of the state-of-the-art LVLM models before fine-tuning: mPLUG-Owl Li et al. (2022), LLaVA Liu et al. (2023b) , InstructBLIP Li et al. (2023a). We collected several fine-tuning approaches on the baseline InstructBLIP in the second rows as presented by Zhou et al. (2023).

The results demonstrate that our proposed method consistently outperforms all non-fine-tuning baselines across hallucination metrics. Remarkably, our approach enhances $CHAIR_i$ by +18.9% and $CHAIR_s$ by +41.3% compared to the baseline InstructBLIP and notably surpasses the performance of *Hard Prompting*. Among fine-tuning approaches, we achieved the top ranking on $CHAIR_s$ and second place on $CHAIR_i$, with a very marginal difference compared to the best-performing model, LURE.

Additionally, our method is able to maintain or enhance caption quality across various metrics. Table 5

| Method | Captioning Quality | | |
|---|---|---|---|
| | *SPICE* (%) ↑ | *BLEU* (%) ↑ | *BERTScore* (%) ↑ |
| mPLUG-Owl | 12.5 | 2.7 | 87.40 |
| LLaVA | 13.5 | 3.0 | 87.83 |
| InstructBLIP | 10.9 | 1.1 | 85.81 |
| Hard Prompt | 11.1 | 1.0 | 85.9 |
| Our | 11.0 | 1.5 | 86.86 |
| Our-Enhance | **14.6** | **6.6** | **90.42** |

Figure 5: Captioning quality is evaluated using NLP metrics, comparing our approach to other methods. **Our** uses only *F1Score* and KL divergence, while **Our-Enhance** incorporates additional metrics: Meteor and BERTScore.

presents our results. The row for **Our** demonstrates the use of *F1Score* and KL divergence, maintaining performance comparable to the base model, InstructBLIP. There is a slight increase in *SPICE*, *BLUE*, and *BERTScore*, which we attribute to the generated captions being more factual, concise, and focused, resulting in shorter and more precise outputs. When ground truth captions are available, incorporating Meteor and BERTScore, as in **Our-Enhance**, significantly improves caption quality. It is evident that **Our-Enhance** significantly improves captioning performance across *SPICE*, *BLEU*, and *BERTScore*, surpassing all previous baselines.

**Extend evaluations to complex dataset:** We conducted additional evaluations using the Visual Genome dataset and the CCEval metric as outlined in Halle-switch Zhai et al. (2023). These evaluations allowed us to explore the model's performance in more complex scenarios, where captions typically contain a denser array of objects, potentially increasing the likelihood of hallucination. The result is shown in Fig. 6.

| Model | $CCEVal_i$ ↓ | $CCEval_s$ ↓ |
|---|---|---|
| LLAVA7B | 72.00 | 19.7 |
| LLaVA13B | 79.00 | 23.8 |
| InstructBlip7B | 72.00 | 22.30 |
| Our | **27.0** | **19.6** |

Figure 6: The performance of our method on the Genome dataset.

Interestingly, the LLaVa13B model, despite being a stronger generative model, shows more hallucinations in both CCEVal-i and CCEVal-s scores compared to LLaVa7B. Examining the generated captions shows that this is due to LLaVa13B's tendency to generate more imaginative content, indicating that while increased model capability can enhance creativity, it may also lead to more hallucinations. Therefore, guiding the model to prioritize factual accuracy is essential.

Fig. 6 also clearly shows the effectiveness of our model in reducing object hallucinations, with significantly lower $CCEVal_i$ (object-level) score and achieve the best $CCEVal_s$ (caption-level) score among baseline models. Although the improvement in $CCEVal_s$ is marginal, this is likely due to the higher object density in Visual Genome Images, which increases the risk of hallucination and makes it more challenging to eliminate hallucinations entirely. Nonetheless, our model demonstrates robustness and adaptability in handling complex captioning tasks, confirming its effectiveness beyond the COCO dataset.

### 4.3 ABLATION STUDY

**Effectiveness of *F1Score*:** The *F1Score* plays a crucial role in ensuring the recall of generated captions. Fig. 7 provides a comparison between using the *F1Score* instead of *CHAIR* in the reward. It is evident that employing *CHAIR* directly has a detrimental effect, significantly re-

| Reward | *Pre* (%) ↑ | *Rec* (%) ↑ | $CHAIR_i$ (%) ↓ | $CHAIR_s$ (%) ↓ |
|---|---|---|---|---|
| Base | 72.9 | 71.3 | 27.1 | 60.9 |
| CHAIR | **93.7** | 20.6 | **6.3** | **14.4** |
| F1Score | 93.2 | **70.2** | 6.8 | 18.8 |

Figure 7: Comparison of *Precision* ($Pre$) and *Recall* ($Rec$) between using *CHAIR* and *F1Score* in the reward function.

ducing the recall. This outcome can be attributed to the sole emphasis on precision without due consideration for recall. The *F1Score* addresses this issue by incentivizing the model to maintain high recall, thus preserving a comprehensive coverage of ground truth objects.

**Ablation on Incorporating NLP Metrics:** Fig. 8 illustrates the impact of using different automatic metrics. The baseline model shows high object hallucination with 25.8% under $CHAIR_i$. Incorporating the *F1Score* significantly reduces hallucination down to 6.8% while maintaining comparable *BERTScore* and *BLEU* score to the baseline. Adding *BERTScore* and *Meteor* to the reward function further enhances caption quality, achieving 92.42 in *BERTScore* and 6.6 in *BLEU* on the COCO test dataset. This ablation study highlights the effectiveness of each component, particularly the *F1Score*'s role in reducing hallucination, and the additional benefits of *BERTScore* and *Meteor* for optimizing caption quality when reference captions are available.

| Base | F1Score | BERTScore | Meteor | $CHAIR_i$ | BERTScore | BLUE |
|---|---|---|---|---|---|---|
| ✓ | | | | 25.8 | 85.81 | 1.1 |
| ✓ | ✓ | | | **6.8** | 86.86 | 1.5 |
| ✓ | ✓ | ✓ | | 6.9 | 90.51 | 1.8 |
| ✓ | ✓ | ✓ | ✓ | 6.9 | **90.42** | **6.6** |

Figure 8: The ablation studies examining the impact of BERTscore and Meteor metrics on the COCO test set

## 5 DISCUSSION

**On the scalability and Computational Resources:** Our framework performs LVLM fine-tuning by leveraging automatic NLP metrics, significantly reducing the reliance on human effort, thus enhancing scalability. The quality of the fine-tuned model depends on automatic metrics like *F1Score*. As more advanced hallucination metrics are developed, our framework can easily integrate them without major changes.

During development, we recognized the significant GPU demands of fine-tuning LVLMs. To address this, we designed the framework with efficiency at its core, eliminating network duplication and leveraging the PEFT approach. It is worth noting that combining mixed precision Micikevicius et al. (2017) with efficient attention mechanisms (e.g. xformers Lefaudeux et al. (2022)) and advanced distributed training methods (e.g. Accelerate Gugger et al. (2022)) synergistically supports our framework's implementation. With adequate GPU resources, our approach is highly suitable. However, future work could explore prediction-time adaptations, such as prompt engineering, to scale models even larger and provide more accessibility to hobbyist researchers. Larger LVLMs, with stronger prompt-following capabilities, are especially likely to benefit from these methods.

**On the detailed caption length:** Our results demonstrate a significant reduction in hallucinations, but we observed a minor side effect: the average caption length is shorter than the baseline (85 tokens compared to 110 tokens). A closer examination revealed that the model's emphasis on factual content leads to the omission of imaginative elements, resulting in shorter captions. Our experiments indicate that penalizing shorter captions (in the reward function) can increase their length to approximately 105 tokens. Unfortunately, this adjustment also raises the hallucination rate to 7.8%. This suggests a trade-off between caption length and hallucination rates that we should be aware of. Balancing these factors is crucial for optimizing performance based on specific needs.

## 6 CONCLUSION

In conclusion, this paper addresses the persistent challenge of object hallucination in LVLMs for image captioning, especially in detailed descriptions. Traditional fine-tuning methods, while effective, face scalability issues due to substantial human effort requirements. To overcome this, we propose a novel framework that leverages reinforcement learning (RL) with automatic natural language processing metrics within an MDP framework. This approach minimizes object hallucination while preserving caption quality, achieved through careful architectural design and a tailored reward function. Our framework effectively reduces hallucination compared to the baseline model, InstructBLIP, while maintaining consistent object coverage and caption quality. With its emphasis on speed and memory efficiency, the framework offers practical scalability and represents a significant advancement in improving the reliability of LVLMs for image captioning.

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

## A  OPEN-VOCABULARY BENCHMARK

In our approach, we evaluate both a closed dataset (COCO) and an open-vocabulary dataset (Visual Genome). For COCO, we selected CHAIR due to its tailored design for this dataset, ensuring reliable and consistent results. For Visual Genome, we opted for CCEVAL, which builds on CHAIR's methodology by incorporating large language models (LLMs) to better capture objects in open-vocabulary settings, particularly in the context of the Visual Genome dataset. Notably, open-vocabulary benchmarks can also be leveraged to evaluate the framework in broader applications.

Specifically, the study Mitigating Open-Vocabulary Caption Hallucinations introduces the Open-Chair benchmark, an extension of CHAIR that accommodates a broader object vocabulary than COCO. OpenChair proposes an evaluation method using LLMs to identify hallucinated objects, providing complementary insights for experiments beyond the COCO dataset. Similarly, ALOHa highlights CHAIR's limitations due to its reliance on string matching for a fixed object set. While CHAIR performs well for COCO, its applicability is limited in open-vocabulary contexts. To overcome this, ALOHa employs LLMs to detect objects in more general settings, enhancing its adaptability.

It is important to note that CCEVAL, OpenChair, and ALOHa all address the limitations of CHAIR and converge on a shared approach: leveraging LLMs to enable more generalized and versatile applications across diverse datasets.

## B  MOTIVATION OF USING REINFORCEMENT LEARNING

Our motivation for employing RL stems from the need to minimize human effort while ensuring effectively reduct hallucination.

Traditional approaches to mitigating hallucinations often require identifying specific sources of hallucination and designing targeted strategies to counter them. While effective, these methods are labor-intensive. Data-driven alternatives like supervised learning provide some level of automation but rely heavily on labeled datasets, which still require significant human input for data annotation and curation—an increasingly costly and time-intensive process, particularly for large-scale models.

In contrast, reinforcement learning in the literature not only demonstrates strong alignment capabilities for LVLMs in tasks like image captioning but also offers a promising path to automation by significantly reducing the need for explicit labels (e.g., relying only on simple binary feedback for reward modeling). We are motivated to push this approach to its limits by completely eliminating human-labeled data, fully leveraging RL's potential through the exclusive use of automatic metrics to reduce hallucinations. These metrics are gradually improving in their alignment with human feedback in terms of both accuracy and reliability. Our approach enables the model to iteratively refine its outputs based solely on automatic feedback, providing an efficient and scalable solution that aligns with the trend toward larger LVLMs.

## C  DESCRIPTIONS OF EVALUATION METRICS

**BLEU:** BLEU (Bilingual Evaluation Understudy) is a metric employed for assessing the quality of machine-generated translations by comparing them to one or more reference translations. Derived from the concept of precision in n-grams—consecutive sequences of n words—BLEU quantifies the extent to which the generated translation aligns with the reference translations in terms of n-gram overlap Papineni et al. (2001)

**BERTScore:** BERTScore is a technique designed to assess the performance of natural language generation or summarization systems, as introduced by Zhang et al. (2019). This method gauges the similarity between a reference text and a generated text by leveraging contextualized embeddings derived from BERT (Bidirectional Encoder Representations from Transformers).

**SPICE:** SPICE (Semantic Propositional Image Caption Evaluation) Anderson et al. (2016) is employed to assess the quality of image captions by evaluating both the semantic content and precision of the generated captions in comparison to reference captions. This metric operates on the hypothesis that semantic propositional content plays a crucial role in human caption evaluation. SPICE introduces an automated caption evaluation method defined over scene graphs, aiming to capture the intricacies of semantic representation in image captions.

**METEOR:** METEOR (Metric for Evaluation of Translation with Explicit ORdering) Banerjee & Lavie (2005) serves as an evaluation metric for machine translation output. This metric calculates the harmonic mean of unigram precision and recall, with recall carrying greater weight than precision. Unlike other metrics, METEOR incorporates additional features such as stemming and synonymy matching, in addition to the standard exact word matching. Its design addresses certain issues identified in the widely used BLEU metric, aiming to improve correlation with human judgment at the sentence or segment level. Notably, METEOR focuses on sentence-level correlation, diverging from BLEU, which seeks correlation at the corpus level.

## D  LARGE VISION-LANGUAGE MODEL

In this paper, the term Large Vision-Language Models (LVLMs) refers to deep learning models designed to process joint visual and textual data, built upon foundational LLMs. Specifically, LVLMs integrate robust Large Language Models (LLMs) with pre-trained Vision encoders to enhance accuracy in understanding and generating language and vision-related content.

Typically, an LVLM is comprised of a vision encoder, a language encoder (i.e., an LLM), and a cross-modal alignment network. The training process for LVLMs involves three primary stages. Initially, the vision and language encoders undergo pre-training on extensive unimodal datasets, focusing on image and text data separately. Subsequently, these encoders are aligned through pre-training on image-text alignment, enabling the LLM to generate meaningful texts corresponding to given images. Finally, the whole model undergoes further fine-tuning on image-text instructions, enhancing its ability to provide satisfactory responses to natural language queries related to specific images. Notably, during the second and third stages, selective fine-tuning of individual components can be performed instead of conducting comprehensive parameter adjustments.

Once the visual encoder and the LLM are effectively aligned, the resulting LVLM exhibits superior visual comprehension capabilities. It not only captures the visual semantics of objects within an image but also delves into linguistic semantics by leveraging the parametric knowledge embedded

in the LLM, achieving enhanced performance across various vision language tasks, such as image captioning.

## E  OBJECT HALLUCINATION AND CHAIR METRICS

**Object Hallucination:**  In literature, the term "object hallucination" denotes a phenomenon wherein a model generates descriptions or captions containing objects that are either inconsistent with or entirely absent from the target image. Object hallucination can be understood and defined at various semantic levels. At its simplest, it pertains to discrepancies at the object level, though more nuanced interpretations may extend to the attributes or characteristics of objects. This study focuses on object-level object hallucinations within model-generated captions, deferring finer-grained analyses of object hallucinations—such as those related to quantity, attributes, and positions—to future investigations.

**CHAIR:**  The Caption Hallucination Assessment with Image Relevance (CHAIR) Rohrbach et al. (2018a) stands as a widely recognized standard for gauging the occurrence of object hallucination in image captioning tasks. This metric operates by scrutinizing the actual objects depicted in an image and subsequently determining the percentage of referenced objects in the generated caption that do not correspond to objects within the image itself. Two distinct variants of CHAIR are employed to measure object hallucination: $CHAIR_s$, which evaluates object hallucination at the caption level, and $CHAIR_i$, which assesses object hallucination at the object level. Mathematically, the metrics are defined as follows:

$$CHAIR_i = \frac{\# \{\text{hallucinated objects}\}}{\# \{\text{all objects in prediction}\}} \tag{12}$$

$$CHAIR_s = \frac{\# \{\text{ captions with hallucinated objects }\}}{\# \{\text{all captions }\}}. \tag{13}$$

## F  DESCRIPTION OF LVLM MODELS USED AS BASELINE

The evaluated LVLMs basically consist of three parts: a visual encoder, an alignment model, and a large language model. All the above models have been tuned on collected visual instruction data

**mPlug-Owl** mPLUG-Owl Ye et al. (2023), is a novel training method that enhances LLMs with multi-modal capabilities by integrating foundational LLM training, a visual knowledge module, and a visual abstractor module. This approach supports various modalities and enhances both unimodal and multimodal abilities through collaborative learning. mPLUG-Owl employs a two-stage training process to align image and text data, leveraging LLM assistance while preserving and enhancing its generative capacities. Initially, the visual knowledge and abstractor modules are trained using a fixed LLM module to align image-text pairs. Subsequently, language-only and multi-modal supervised datasets are utilized to fine-tune a Low-Rank Adaptation (LoRA) module on LLM and the abstractor module while keeping the visual knowledge module frozen.

**LLaVA** uses a linear projector to map visual token as a soft-prompt into LLM input tokens. LLaVA has a two-stage training, where the initial stage focuses on simple caption pretraining solely for the linear projector, while the subsequent stage finetunes both the projector and LLM on instruction data. Instruction data leverages language-only GPT-4 by inputting visual ground truth from COCO dataset.

**InstructBLIP** adopts the BLIP-2 architecture, and is distinguished by its training of a Q-former, which bridges the frozen vision encoder and LLM. InstructBLIP's instruction fine-tuning spans across 26 distinct datasets.

## G  INSTRUCTION TEMPLATE FOR DETAILED IMAGE CAPTIONING IN COCO DATASET

We use Instruction Templates to generate long, detailed captions. During training, the prompt is randomly selected to query the LVLM. The Instruction Templates are at below:

- ⟨Image⟩A detailed image caption:
- ⟨Image⟩A detailed image description:
- ⟨Image⟩Write a long description for the image.
- ⟨Image⟩Describe the content of the image in detail.
- ⟨Image⟩Can you explain clearly what you see in the image?
- ⟨Image⟩Could you describe clearly what you perceive in the photo?
- ⟨Image⟩Please provide a detailed depiction of the picture.
- ⟨Image⟩Provide a detailed description of the given image.

## H  HARD PROMPT DESIGN

We have developed a set of "hard prompts" intended to be appended at the beginning of the input instruction, aiming to mitigate object hallucination in the model's generated captions. Each prompt is meticulously crafted to target specific sources of object hallucination, strategically guiding the model away from potential pitfalls. Below is the comprehensive list of prompts:

- Directly prohibit object hallucination : "Please don't hallucinate the objects in the image"
- Emphasize concrete details: "Provide captions based on specific, easily identifiable elements in the image."
- Prioritize realism: "Generate captions that reflect plausible scenarios and avoid fantastical or improbable elements."
- Stick to visible entities: "Describe only what is clearly visible in the image and avoid making assumptions about hidden or obscured objects."
- Be conservative in interpretation: "Refrain from extrapolating beyond what is evident in the image; captions should stay closely tied to observable elements."
- Avoid creative interpretations: "Discourage the generation of captions that involve imaginative or metaphorical representations of the scene."
- Limit descriptive scope: "Keep captions focused on the central objects or subjects in the image, avoiding unnecessary details or peripheral elements."
- Minimize speculative language: "Generate captions with certainty, avoiding speculative language or uncertain descriptions of the depicted scene."
- Resist contextual speculation: "Do not create captions that rely on external context or background information not present in the image."
- Steer clear of abstract concepts: "Refrain from incorporating abstract or conceptual ideas into the captions; stick to tangible, visible elements."
- Encourage literal language: "Favor literal and straightforward language in captions, avoiding figurative expressions or interpretations."

## I  DETAILED ABOUT PROMPT TUNNING

Image captioning with the Large Vision Language Model (LVLM) represents a crucial text generation task. Departing from the traditional classification approach, which assesses the probability of an output class given input as $P(y|X, Z)$, where X comprises tokens representing the instruction, y denotes a single class label, and Z contains tokens representing an image, we now adopt a conditional generation perspective. In this paradigm, Y signifies a sequence of tokens that form a caption. The captioning process by Large Vision Language Models is expressed as $P_\theta(Y|X, Z)$, where $\theta$ represents the model's weights.

Prompting involves augmenting the model's generation of Y by providing additional context for it to rely on. This is achieved by prefixing a sequence of tokens, G, denoted as $\{g_1, g_2, ..., g_k\}$, to the input X, such that enabling the model to enhance the likelihood of generating the correct Y: $P_\theta(Y|[G; X], Z)$. Throughout this process, the model parameters, $\theta$, remain unchanged. Optimal

prompt selection can be achieved through manual exploration of prompt tokens, known as *Hard Prompting*, or by representing G with dedicated parameters, $\phi$, which model the embeddings of these tokens. These parameters are then refined using gradient descent. This technique is termed *Soft Prompting*. Consequently, our updated conditional generation is expressed as $P_{\theta;\phi}(Y|[G;X],Z)$, and it can be trained by maximizing the reward through backpropagation, with gradient updates solely applied to $\phi$.

The modeling of Soft Prompting is straightforward. When presented with a sequence of $n$ tokens, $\{x_1, x_2, \ldots, x_n\}$, the initial step undertaken by LVLM involves embedding these tokens to create a matrix $X_e \in \mathbb{R}^{n \times e}$, where $e$ denotes the dimension of the embedding space. Our soft prompts are expressed as a parameter $G_e \in \mathbb{R}^{k \times e}$, with $k$ being the length of the prompt. Subsequently, the prompt is concatenated to the embedded input, resulting in a unified matrix $[G_e; X_e] \in \mathbb{R}^{(k+n) \times e}$, which is then processed through the LVLM as per usual. During training, our models are designed to maximize the return of Y. However, it is noteworthy that only the prompt parameters $G_e$ undergo updates, ensuring the model learns to effectively utilize the provided prompts while keeping other parameters fixed.

## J  DATASET DESCRIPTION

**Visual Genome** contains Visual Question Answering data in a multi-choice setting. It consists of 101,174 images from MSCOCO with 1.7 million QA pairs, 17 questions per image on average. Compared to the Visual Question Answering dataset, Visual Genome represents a more balanced distribution over 6 question types: What, Where, When, Who, Why and How. The Visual Genome dataset also presents 108K images with densely annotated objects, attributes and relationships.

**The MS COCO (Microsoft Common Objects in Context) dataset** is a large-scale object detection, segmentation, key-point detection, and captioning dataset. The dataset consists of 328K images.

Splits: The first version of MS COCO dataset was released in 2014. It contains 164K images split into training (83K), validation (41K) and test (41K) sets. In 2015 additional test set of 81K images was released, including all the previous test images and 40K new images.

Based on community feedback, in 2017 the training/validation split was changed from 83K/41K to 118K/5K. The new split uses the same images and annotations. The 2017 test set is a subset of 41K images of the 2015 test set. Additionally, the 2017 release contains a new unannotated dataset of 123K images.

The dataset has annotations for:

- object detection: bounding boxes and per-instance segmentation masks with 80 object categories.
- captioning: natural language descriptions of the images.
- keypoints detection: containing more than 200,000 images and 250,000 person instances labeled with keypoints (17 possible keypoints, such as left eye, nose, right hip, right ankle).
- stuff image segmentation: per-pixel segmentation masks with 91 stuff categories, such as grass, wall, sky.
- panoptic: full scene segmentation, with 80 thing categories (such as person, bicycle, elephant) and a subset of 91 stuff categories (grass, sky, road).
- dense pose: more than 39,000 images and 56,000 person instances labeled with DensePose annotations – each labeled person is annotated with an instance id and a mapping between image pixels that belong to that person body and a template 3D model. The annotations are publicly available only for training and validation images.

## K  QUANLITATIVE RESULT

We present quantitative results on the COCO dataset. The Base refers to the InstructBlip model, while Our model is the fine-tuned version of InstructBlip using our framework. Hallucinated objects are highlighted in bold and red for clarity.

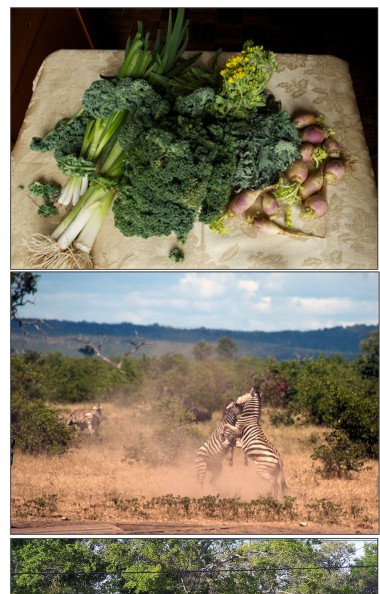

**Base:** The image displays a collection of various vegetables and greens, all neatly arranged on top an old-fashioned tablecloth. There are multiple rows with different types of produce: some have white ends while others possess more green coloration throughout the stalks or leaves. Various **knives** can be seen nearby in case they're needed for cutting up any larger portions during preparations later downstream from this setup stage involving setting out fresh veggies to drying/dehydrating them by placing those already cut into **glass jars** also known as "pickling"....

**Our:** A dining table displays a vibrant array of fresh vegetables, including both leafy greens and root veggies. Various colors like purple-tinged kale are present among the organic produce on display next to each other. In total, there are about 10 different kinds of plants arranged in an attractive composition…

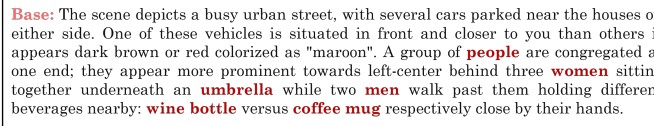

**Base:** The image showcases a pair of zebras playfully fighting in the grass. They may be playing together rather than engaging in territorial battles commonly associated between wild animals such as **lions** vs **hyenas** which usually happen for food sources only.

**Our:** In this image, several zebras are interacting with each other in a lush green field. Some zebras are standing on top of each other, while others are running around the field. There are several trees visible in the background, adding to the natural setting.

**Base:** The scene depicts a busy urban street, with several cars parked near the houses on either side. One of these vehicles is situated in front and closer to you than others it appears dark brown or red colorized as "maroon". A group of **people** are congregated at one end; they appear more prominent towards left-center behind three **women** sitting together underneath an **umbrella** while two **men** walk past them holding different beverages nearby: **wine bottle** versus **coffee mug** respectively close by their hands.

**Our:** In this image, a red van is parked on the side of a road in front of a house. The van is situated near a stop sign, which is located close to the front of the house. There are several trees visible in the background, adding to the scenic setting.

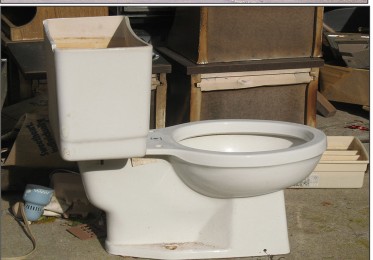

**Base:** The image features a toilet bowl placed on the ground near an old **couch**. Next, there are several stacks of boxes and bags surrounding it. A **television** is situated nearby with its base visible towards left side next section by **stairs** leading downward likely used during restoring activities.

**Our:** In this image, there is a white toilet sitting on the ground in a junkyard. The toilet appears to be disassembled, with some of its parts scattered around the area.

## L    TRAINING SETTINGS FOR PROMPT FINE-TUNING

We present parameter settings in our framework during the training phase on the COCO dataset. Here we use 8 A6000 50G GPU for training. The information is shown in table below:

| Parameters | |
|---|---|
| Max Soft Prompt Length $n$ | 20 |
| Max Episode Length $H$ | 150 |
| Mini-Batch Size | 64 |
| PPO Learning Rate | 0.0002 |
| Kullback–Leibler (KL) Initial Coefficient $\lambda$ | 0.1 |
| KL Target Coefficient | 0.05 |
| Number of Training Epochs | 50 |
| Reward Weight for Meteor $\alpha$ | 0.1 |
| Reward Weight for BERTScore $\beta$ | 1 |
| Vocabulary Size $\mathcal{V}$ | 32000 |

Table 2: Training parameters of our framework for the COCO dataset

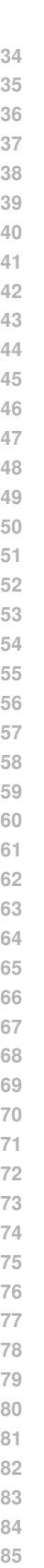

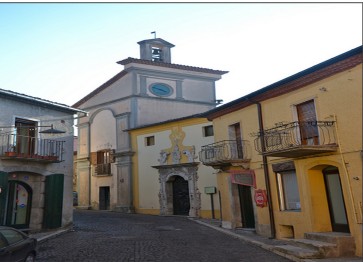

**Base:** The image depicts a small alleyway lined with buildings and balconies, featuring several **bikes** parked in the space between them. There are numerous **motorcycles** placed throughout this area. Additionally, there is another object that appears like it might be either one or two **cars** behind some nearby **trees** towards both sides near various parking spaces along city street setting.

**Our:** In this scenic view, a narrow cobblestone street winds its way through a charming old town. The street is lined with various buildings, some of which have multiple floors. There are several cars parked along the street, adding to the vibrant atmosphere.

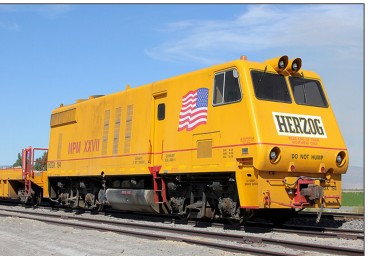

**Base:** The image depicts a train moving along the railway tracks, with an open door on one of its cars. On either side there are multiple **smaller vehicles** parked nearby but do not appear involved into transportation as they resemble passenger automobiles without any visible loads travelers would board them at stops like passengers embark trains around station entrances while loading/unloading goods carried onto flatbed utility wagons (with wheels).

**Our:** At the train station, a large yellow train is parked on the tracks, waiting for its next departure. The train is decorated with an American flag, adding a patriotic touch to the scene.

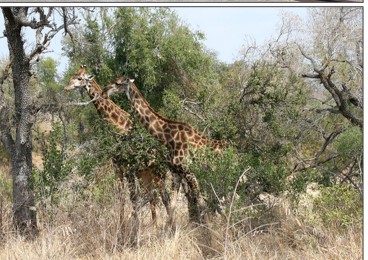

**Base:** The image captures a pair of giraffes in their natural habitat, roaming across the forest. They are standing and walking among bushy trees covered with various leaves surround them like an umbrella canopy covering parts of this jungle landscape scene near where they live. There is at least three **people** present throughout the environment: one **person** sitting right next to another close by; several others scattered around both left-side side areas.

**Our:** In this scenic image, two giraffes are standing in a lush, green forest. They appear to be socializing with each other while browsing through the trees. The giraffes are spread out across the forest, with one giraffe standing closer to the left side of the image, and the other further to the right.

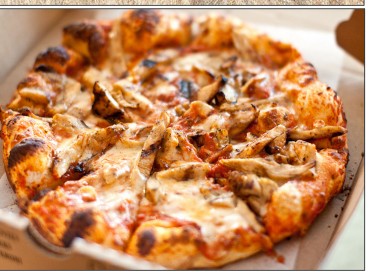

**Base:** The image features a large pizza with various toppings in an open box. Two empty **bowls** can be seen near or underneath one side of another tabletop located offscreen towards left behind them. A **person** stands next to these tables having dinner together at some point during their meal preparation session where they enjoyed eating both savory pasta dishes as well as indulging themselves.

**Our:** A delicious mushroom pizza is sitting in a box, ready to be enjoyed. The pizza is sliced into multiple pieces, making it easy to share or enjoy on its own.

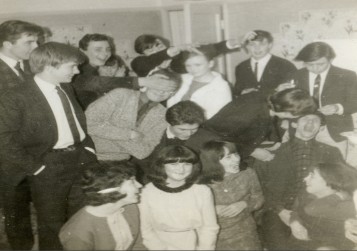

**Base:** The image depicts a group of young people gathered together in an old-fashioned setting. They are wearing various clothing and ties, with some sporting bowties or traditional dresses. A room is visible behind them as well featuring several **tables** set up inside it where guests may have been seated. Furthermore, there appears two **clock radios** placed towards different corners within that space giving evidence of activity happening here throughout time frames.

**Our:** In this vintage photograph, a large group of people are gathered together in a room. They are all dressed in suits and ties, creating a formal atmosphere. The group is engaged in socializing and interacting with each other.

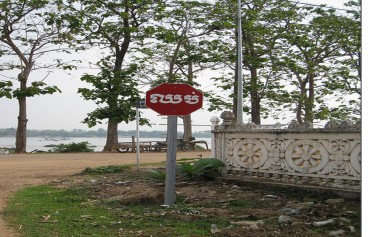

**Base:** The image captures a beautiful day with trees in the background and **people** walking around. There is an area of dirt on one side near some water where three individuals can be seen engaging together or resting underneath two **umbrellas.** Several park **benches** are present along to each other as well towards both sides closer by the ground surface itself containing four seating places among them at least six **chairs** have been positioned further off but still easily visible through such foliation.

**Our:** In this scenic image, there is a red stop sign standing tall in the middle of a dirt road next to a fence. The sign is placed near a motorcycle, which can be seen in the background.

1188
1189
1190
1191
1192
1193
1194
1195
1196
1197
1198
1199
1200
1201
1202
1203
1204
1205
1206
1207
1208
1209
1210
1211
1212
1213
1214
1215
1216
1217
1218
1219
1220
1221
1222
1223
1224
1225
1226
1227
1228
1229
1230
1231
1232
1233
1234
1235
1236
1237
1238
1239
1240
1241

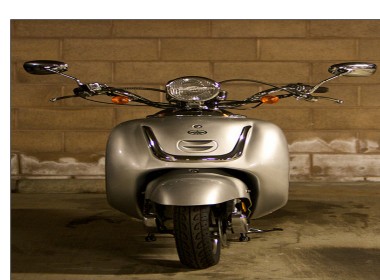

**Base:** The image showcases a shiny silver moped, parked neatly inside of an underground garage. There are two rear wheels on either side that make up most part of this compact vehicle's frame area near its tail section. A few **people** can be seen walking within various areas throughout the scene - specifically between right middle (one **person**), top centralized portion just beyond three riders sitting there beside another standing individual present alongside several **vehicles** also situated across four main locations.

**Our:** In this image, a sleek and modern motor scooter is parked in front of a brick wall. The motor scooter is silver in color and appears to be well-maintained. There are several motorcycles visible in the scene, creating a vibrant and lively atmosphere.

