# OpenReview forum: "Mitigating Object Hallucination in Large Vision Language Model with Human-Free Reinforcement Learning"
_ICLR.cc/2025/Conference — Submitted to ICLR 2025_

### Official Review · Reviewer_9y2t · 2024-10-26

**Soundness:** 2
**Presentation:** 3
**Contribution:** 2
**Rating:** 5
**Confidence:** 4

**Summary:**

This paper proposes a human-free RL alignment framework to mitigate object hallucinations in multimodal LLMs, where the output captions contain non-existent objects. The aim of this paper is to relieve the burden of data collection and label annotation for fine-tuning or performing RLHF. To achieve this goal, they proposed utilizing the F1 score together with KL divergence as the reward and then applying PPO for model alignment. More specifically, the F1 score encourages the output captions with fewer hallucinations, while the KL divergence maintains the quality of image captions from the base model. The experiments and quantitative comparisons are conducted on the commonly used COCO dataset.

**Strengths:**

1. Alleviating object hallucination issues is crucial for trustworthy model deployment in several real-world applications, such as healthcare or manufacturing. This task aims to improve the faithfulness of multimodal LLMs on image captioning tasks, which could benefit image captioning tasks requiring high standard truthfulness, like report generation from medical images.
2. The proposed reward function constructed by the F1 score and KL divergence is reasonable and technical sound.
3. The overall paper is easy to follow.

**Weaknesses:**

1. This paper trains and evaluates the models on the COCO dataset. While this training manner avoids additional effort for collecting and annotating human preference data, I’m wondering whether the original capabilities of LVLMs (i.e., base model) can be adequately preserved. This paper only demonstrates the results of captioning quality on the COCO dataset in Figure 5. However, does the tuned model still maintain the captioning capability of the base model compared to other datasets? My concern is whether the original generalizability of the base model would be destroyed after being tuned by the proposed method.
2. In addition, the title of this paper claims to mitigate object hallucinations in large vision language models. Can other vision-language tasks like VQA or visual chatting also benefit from the proposed RL tuning process? Only considering image captioning tasks cannot fully support the claim of mitigating object hallucinations of LVLMs.
3. From the qualitative results shown in Section I of the Appendix, the outputs from the proposed method are shorter than those of the base model. As we know, the outputs from LVLMs are generally more detailed and comprehensive than the ground truth of the COCO dataset. The results in the Appendix show that the detailed captioning properties of LVLMs are destroyed. More clarifications are encouraged.

**Questions:**

Please refer to the Weaknesses. The following is a minor question.

1. This paper only considers InstructBLIP to be the LVLM. Are other commonly used LVLMs, such as LLaVA or VILA, also applicable to this proposed RL tuning framework?

---

> ### Author Response · Authors · 2024-11-24
> **Author's Response**
>
> We thank the reviewer for the constructive feedback and provide our response as follows:
>
> **Weaknesses:**
>
> > 1. This paper trains and evaluates the models on the COCO dataset. While this training manner avoids additional effort for collecting and annotating human preference data, I’m wondering whether the original capabilities of LVLMs (i.e., base model) can be adequately preserved. This paper only demonstrates the results of captioning quality on the COCO dataset in Figure 5. However, does the tuned model still maintain the captioning capability of the base model compared to other datasets? My concern is whether the original generalizability of the base model would be destroyed after being tuned by the proposed method.
>
> Thank you for your insightful concern. The preservation of captioning quality is ensured by the use of KL divergence in our framework. KL divergence helps maintain alignment with the baseline model (referred to as the reference model in our framework), preventing the tuned model from deviating too much from the original output. Without this mechanism, training could prioritize specific metrics like F1 score, potentially disrupting the caption structure, as we observed in our initial experiments without KL divergence.

---

> ### Author Response · Authors · 2024-11-24
> **Author's Response 1**
>
> > 2. In addition, the title of this paper claims to mitigate object hallucinations in large vision language models. Can other vision-language tasks like VQA or visual chatting also benefit from the proposed RL tuning process? Only considering image captioning tasks cannot fully support the claim of mitigating object hallucinations of LVLMs.
>
> In this paper, we focus on image captioning to provide a more in-depth analysis, leaving other tasks for future investigation. To answer the reviewer’s question, our reinforcement learning framework is flexible and can work with different queries, not limited to captioning or Q&A, without specific adjustments. However, evaluating its effectiveness on tasks beyond captioning is outside this paper’s scope.
>
> As a preliminary test, we adapted the COCO dataset to a VQA task with the prompt, "What is in the given image?" Our model achieved CHAIRi and CHAIRs scores of 7.9 and 20.5, respectively, compared to instructBLIP's 22.8 and 40.1. This indicates potential but suggests further exploration is needed for VQA tasks.
>
> > 3.  From the qualitative results shown in Section I of the Appendix, the outputs from the proposed method are shorter than those of the base model. As we know, the outputs from LVLMs are generally more detailed and comprehensive than the ground truth of the COCO dataset. The results in the Appendix show that the detailed captioning properties of LVLMs are destroyed. More clarifications are encouraged.
>
> It is worth noting that even the shortened captions generated by our method remain significantly longer than the ground-truth captions. The average lengths are 85, 110, and 10 for our method, the baseline, and the ground truth, respectively. This concern is similar to Reviewer 2’s comment, and we briefly address it in the Discussion section, "On the detailed caption length" (lines 519–520). To clarify further, we provide a detailed explanation here.
>
> The proposed framework under reinforcement learning (RL) optimizes the output to satisfy the reward function, which may implicitly result in shorter captions. However, explicit constraints are in place to prevent undesirable behavior. These include preserving object coverage (the Recall component of the F1 Score) and ensuring that the KL divergence from the original output does not deviate excessively. These constraints ensure that the method does not simply shorten captions arbitrarily but focuses on improving object-mention accuracy (the Precision component of the F1 Score) and coverage. Therefore, the shortening effect is merely a side effect of optimization which the primary goals can dominate.
>
> As discussed in the Discussion section, checking the actual outputs reveals that, while the captions are shorter, they maintain factual accuracy and eliminate imaginative or hallucinatory content. We refer the reviewer to the qualitative results in the Appendix for further validation. We argue that this shortening effect is not necessarily negative, as it offers a viable alternative: a concise, accurate caption may be preferable to a longer one with imaginative details, depending on user needs.
>
> Additionally, we provide an option to encourage the model to generate longer captions by incorporating a length component into the reward function (lines 523- 525). This adjustment results in slightly longer captions, with a modest decrease in hallucination reduction. However, the hallucination rate remains significantly lower compared to the baseline, demonstrating the flexibility of the proposed framework.
>
> **Question**
>
> > 1.  This paper only considers InstructBLIP to be the LVLM. Are other commonly used LVLMs, such as LLaVA or VILA, also applicable to this proposed RL tuning framework?
>
> Yes, our method is agnostic to the specific large vision-language model (LVLM) used. The framework simply augments the model by prefixing a soft prompt to the query and adding a linear layer on top of the latest hidden latents. This design is compatible with and can be easily implemented on any LVLM.
>
> As suggested by another reviewer, we conducted experiments with LLaVA and provide the results below:
>
> | **Model**      | **CHAIRi** | **CHAIRs** |
> |-----------------|------------|------------|
> | **LLaVA-base** | 22.5       | 62.7       |
> | **LLaVA-our**  | 7.7        | 22.2       |
>
> These results demonstrate that our method can be seamlessly integrated into LLaVA without any issues. Other LVLMs, such as VILA, can also be incorporated similarly.

---

> > ### Comment · Reviewer_9y2t · 2024-11-25
> >
> > Thanks to the authors for providing the response. I have read the rebuttal by the authors and comments from other reviewers. The rebuttal, especially for W1 and W2, has not fully convinced me. The response implies that only image captioning tasks can be applicable, but the capability of LVLM to perform other tasks would not be sufficiently preserved. Thus, the title "Mitigating Object Hallucination in Large Vision Language Model" seems to be overclaimed. I will keep my original rating.

---

> > > ### Author Response · Authors · 2024-11-26
> > > **Feedback to the reviewer**
> > >
> > > Thank you for your thoughtful feedback. We greatly appreciate your engagement with our work.
> > >
> > > Regarding your concern about the title, our intention was to maintain a concise and impactful phrasing. While we acknowledge that the title may not explicitly specify the focus on image captioning tasks, we believe that the abstract and the main body of the paper sufficiently clarify the scope and application of our work. The abstract, in particular, explicitly emphasizes that our proposed method is designed and evaluated for image captioning tasks, ensuring that readers understand the intended focus.
> > >
> > > Concerning the statement, “the capability of LVLM to perform other tasks would not be sufficiently preserved,” we respectfully disagree. This was not our intended implication. Our framework is prompt-agnostic and not inherently tied to a specific task such as image captioning. While our experiments primarily focused on captioning tasks, this was due to the constraints of the rebuttal period. However, we anticipate that the framework can generalize effectively to other tasks.
> > >
> > > To address this, we provided preliminary results on a Visual Question Answering (VQA) in W2, demonstrating a significant reduction in hallucination, which underscores the broader applicability of our approach. While these results are initial, they reinforce our belief in the generalizability of the framework to various vision-language tasks. We look forward to exploring this further in future work and providing more comprehensive evidence.
> > >
> > > We hope this addresses your concerns and clarifies the potential of our approach. Thank you again for your valuable input.

---

### Official Review · Reviewer_X6g4 · 2024-11-03

**Soundness:** 2
**Presentation:** 3
**Contribution:** 2
**Rating:** 5
**Confidence:** 3

**Summary:**

This paper addresses the challenge of object hallucination in Large Vision Language Models (LVLMs) for image captioning by introducing a human-free framework that employs reinforcement learning driven solely by automatic natural language processing metrics. Central to this approach is the notion that the image captioning task can be effectively modeled as a Markov Decision Process (MDP), due to its inherently sequential nature, where each token generation represents a decision influenced by the current state. Building on this framework, the paper fine-tunes the existing model using a dedicated reward function alongside Proximal Policy Optimization (PPO). This method achieves performance that is comparable to or even superior to alternative approaches, all without any human involvement.

**Strengths:**

Framing the image captioning task as a Markov Decision Process (MDP) and utilizing reinforcement learning driven exclusively by automatic natural language processing metrics is both reasonable and intriguing.

**Weaknesses:**

1. This paper currently validates the effectiveness of the proposed method only based on InstructBLIP, making it difficult to verify its broader applicability. The method’s generalizability across different baselines, such as mPLUG-Owl and LLaVA, should be demonstrated.
2. The authors use the F1 Score to address object hallucination, but this approach is not well-motivated. While F1 Score balances accuracy and recall, it does not have a direct connection with the issue of object hallucination. The authors should provide more theoretical and experimental analysis to support their rationale. In lines 521-523, they mention a minor side effect of their method: the average caption length is shorter than the baseline (85 tokens compared to 110 tokens). This raises the question of whether their proposed framework reduces object hallucination simply by shortening the output length.
3. Using prompt tuning to reduce object hallucination has indeed shown some effectiveness. However, the prompts used by the authors are automatically generated without human intervention, which means these prompts themselves might contain hallucinations or errors. Thus, evaluating the accuracy and validity of these generated samples becomes a critical issue. The authors should provide additional analysis to demonstrate the reliability of these prompts to better support the effectiveness of this approach.

**Questions:**

See Weakness

---

> ### Author Response · Authors · 2024-11-24
> **Author's Response**
>
> We thank the reviewer for the constructive feedback and provide our response as follows:
>
> **Weaknesses:**
>
> > 1. This paper currently validates the effectiveness of the proposed method only based on InstructBLIP, making it difficult to verify its broader applicability. The method’s generalizability across different baselines, such as mPLUG-Owl and LLaVA, should be demonstrated.
>
> We provide additional experimental result, further applying our method to the base LLaVA model. Training was conducted for 50 epochs using the same hyperparameters as InstructBlip. The results are shown below:
>
> | **Model**      | **CHAIRi** | **CHAIRs** |
> |-----------------|------------|------------|
> | **LLaVA-base** | 22.5       | 62.7       |
> | **LLaVA-our**  | 7.7        | 22.2       |
>
> The results may improve further with careful hyperparameter selection. Nevertheless, these results demonstrate that our method can be effectively applied to a broader range of baseline models.
>
> > 2.1 The authors use the F1 Score to address object hallucination, but this approach is not well-motivated. While F1 Score balances accuracy and recall, it does not have a direct connection with the issue of object hallucination. The authors should provide more theoretical and experimental analysis to support their rationale
>
> The reviewer may have misunderstood the formulation of the F1 Score used in our method. The F1 Score combines precision and recall, where precision is calculated as the ratio of correctly predicted objects to the total number of predicted objects. This can be expressed as:
>
> $$
> \text{Precision} = \frac{\text{correct objects}}{\text{total predicted objects}} = \frac{\text{total predicted objects} - \text{hallucinated objects}}{\text{total predicted objects}} = 1 - \text{CHAIRi}.
> $$
>
> An increase in precision directly correlates with a decrease in the object hallucination rate. Thus, improving precision inherently reduces hallucination, making the use of the F1 Score well-suited for this task.
>
>
> > 2.2  In lines 521-523, they mention a minor side effect of their method: the average caption length is shorter than the baseline (85 tokens compared to 110 tokens). This raises the question of whether their proposed framework reduces object hallucination simply by shortening the output length.
>
> The proposed framework under reinforcement learning (RL) will optimize the output to satisfy the reward function by any means, which may implicitly involve shortening the output length. However, there are explicit constraints that prevent undesirable behavior, such as preserving object coverage (Recall part of the F1 Score) and ensuring that KL divergence does not deviate too far from the original output. These constraints prevent the method from simply shortening the sentence, focusing instead on improving object-mention accuracy and coverage. The shortening is merely a side effect.
>
> As we point out and discuss in the Discussion section, "On the detailed caption length," checking the actual output shows that even though the caption is shorter, it maintains factual content and eliminates imaginative content. We direct the reviewer to the qualitative results in the appendix for further validation. We argue that this shortening effect is not necessarily negative but offers a viable option: a concise, accurate caption may be preferable to a longer one with imaginative details, depending on user needs. It is also worth noting that even the shortened captions generated by our method are still significantly longer than the ground-truth captions. The average lengths are 85, 110, and 10 for our method, the baseline, and the ground truth, respectively.
>
> Furthermore, the authors also provide a way to encourage the model to generate longer captions by incorporating longer captions into the reward function. This results in a slight decrease in hallucination but still shows a significant reduction compared to the baseline.

---

> ### Author Response · Authors · 2024-11-24
> **Author 's Response 1**
>
> > 3. Using prompt tuning to reduce object hallucination has indeed shown some effectiveness. However, the prompts used by the authors are automatically generated without human intervention, which means these prompts themselves might contain hallucinations or errors. Thus, evaluating the accuracy and validity of these generated samples becomes a critical issue. The authors should provide additional analysis to demonstrate the reliability of these prompts to better support the effectiveness of this approach.
>
> We may not have fully captured the reviewer’s concern, but we interpret it as a suggestion that the method may be suboptimal and requires further experimentation to validate its effectiveness. To clarify, the "prompt" in our method, more specifically the “soft prompt,” consists of trainable parameters (not human-readable prompts). These parameters are optimized via reinforcement learning (RL) to guide the model in avoiding hallucinated objects.
>
> Regarding optimization, the effectiveness of this approach—like any RL-based method—depends on how well RL is performed. If RL is executed correctly and converges to an optimal solution, it can generally provide significant reductions in object hallucination across datasets. While suboptimal solutions may still exist (as you pointed out), our experiments show that the optimization process in this paper has been effectively executed. Empirical evidence demonstrates consistently low object hallucination rates across datasets while maintaining good caption quality (e.g., maintaining BLEU, BERTScore, and SPICE metrics).
>
>
> In response to Reviewer 1’s suggestion, we further applied our method to the base LLaVA model. Training was conducted for 50 epochs using the same hyperparameters as InstructBlip. The results are as follow
>
> | **Model**      | **CHAIRi** | **CHAIRs** |
> |-----------------|------------|------------|
> | **LLaVA-base** | 22.5       | 62.7       |
> | **LLaVA-our**  | 7.7        | 22.2       |
>
> The results may improve further with careful hyperparameter selection. Nevertheless, these results demonstrate that our method can be effectively applied to a broader range of baseline models.

---

> ### Author Response · Authors · 2024-12-03
> **Follow-up on Paper Rebuttal**
>
> Thank you for taking the time to review my paper. I have carefully addressed your feedback in the rebuttal, providing clarifications and updates that I believe significantly enhance the paper. As the discussion period approaches its deadline, I kindly remind you to review the feedback and, if appropriate, consider revising your score. Thank you again for your valuable input and consideration.

---

### Official Review · Reviewer_kqoa · 2024-11-04

**Soundness:** 2
**Presentation:** 2
**Contribution:** 2
**Rating:** 5
**Confidence:** 4

**Summary:**

This paper tackles the problem of object hallucination in large vision-language models for image captioning. The key technical contribution is a human-free framework based on reinforcement learning and automatic NLP metrics.  The framework formulates caption generation as a Markov Decision Process, using a reward function that balances hallucination reduction with maintaining caption quality.  Experiments show this approach significantly reduces hallucination while preserving or improving caption quality compared to baselines, without needing any human input.

**Strengths:**

1. This paper proposes a new reinforcement learning method for reducing the object hallucination.

2. This work points out that the existing metric CHAIR ignores the recall and proposes the F1 metric that evaluates both precision and recall.

**Weaknesses:**

1. The following two relevant works are ignored in the paper. Both paper propose new benchmarks for object hallucination.
[A] Ben-Kish et al., Mitigating Open-Vocabulary Caption Hallucinations.
[B] Petryk et al., ALOHa: A New Measure for Hallucination in Captioning Models.

2. The motivation of using reinforcement learning is not clearly explained.

3. The paper relies pretrained an object detector to eliminate human annotations. Therefore, the performance of the model relies on and might be biased to the object detector used. There is no discussion on this point in the paper.

4. More recent baselines should be compared. The baselines (LLAVA, mPLUG-Owl) in Table 4 are from 2023. New versions of LLAVA and mPLUG-Owl are introduced in 2024.

Post-rebuttal

I have read all the author responses and comments from other reviewers. While W1, W2, and W3 are addressed, I am still not convinced about W4. It is a bit suspicious to see that LLaVA-next and LLaVA-Base achieve similar image captioning quality. I think the paper will benefit from a more comprehensive comparison with newer VLMs like LLaVA-Next. I also agree with another reviewer on the misleading title which seems to claim better performance on general VLM tasks. The paper will be enhanced if the proposed method also works for VQA. Therefore, I would keep my original score.

**Questions:**

1. What is the reason of adopting a different metric for the Genome dataset? why not use F1 or CHAIR?

---

> ### Author Response · Authors · 2024-11-24
> **Authors' Response**
>
> We thank the reviewer for the constructive feedback and provide our response as follows:
>
> **Weaknesses:**
>
> > 1. The following two relevant works are ignored in the paper.
>
> We appreciate the reviewers’ suggestions to consider additional cited papers. These studies, which propose valuable benchmarks and corresponding metrics, offer important insights for open-vocabulary evaluation beyond the COCO dataset. We will incorporate them into the related work section to enhance the paper’s comprehensiveness.
>
> Specifically, the study “Mitigating Open-Vocabulary Caption Hallucinations” introduces the OpenChair benchmark, which extends CHAIR to a broader object vocabulary than COCO and suggest a evaluation method using large language models (LLMs) to identify hallucinated objects. This approach is claimed to complement CHAIR for experiments beyond the COCO dataset. Similarly, ALOHa highlights limitations in CHAIR due to its reliance on string matching for a fixed set of objects, making it suitable for COCO but less generalizable to other datasets. To address this, ALOHa employs LLMs to identify objects in open-vocabulary settings, enhancing its adaptability.
>
> In our approach, we evaluate both a closed dataset (COCO) and an open-vocabulary dataset (Visual Genome). For COCO, we selected CHAIR due to its specialization for this dataset, which ensures reliable and consistent results. For Visual Genome, we chose CCEVAL, which extends CHAIR’s methodology by incorporating LLMs to more effectively capture objects in open-vocabulary settings, with a focus on the Visual Genome dataset. It is worth noting that CCEVAL, OpenChair, and ALOHa all identify similar limitations of CHAIR and share a common approach of using LLM support for a more generalized application.
>
> > 2. The motivation of using reinforcement learning is not clearly explained.
>
> We appreciate the feedback regarding our use of reinforcement learning (RL) to reduce hallucinations in large vision-language models (LVLMs). Our motivation for employing RL stems from the need to minimize human effort while ensuring effectively reduct hallucination.
>
> Traditional approaches to mitigating hallucinations often require identifying specific sources of hallucination and designing targeted strategies to counter them. While effective, these methods are labor-intensive. Data-driven alternatives like supervised learning provide some level of automation but rely heavily on labeled datasets, which still require significant human input for data annotation and curation—an increasingly costly and time-intensive process, particularly for large-scale models.
>
> In contrast, reinforcement learning in the literature not only demonstrates strong alignment capabilities for LVLMs in tasks like image captioning but also offers a promising path to automation by significantly reducing the need for explicit labels (e.g., relying only on simple binary feedback for reward modeling). We are motivated to push this approach to its limits by completely eliminating human-labeled data, fully leveraging RL’s potential through the exclusive use of automatic metrics to reduce hallucinations. These metrics are gradually improving in their alignment with human feedback in terms of both accuracy and reliability. Our approach enables the model to iteratively refine its outputs based solely on automatic feedback, providing an efficient and scalable solution that aligns with the trend toward larger LVLMs.

---

> ### Author Response · Authors · 2024-11-24
> **Authors' Response 1**
>
> > 3. The paper relies pretrained an object detector to eliminate human annotations. Therefore, the performance of the model relies on and might be biased to the object detector used. There is no discussion on this point in the paper.
>
> In case no ground true object is available, the performance of hallucination reduction depend on the accuracy of the object detector used. More accurate ground true object extraction result in better hallucination reduction. To provide a sense of how the object detector compares to ground-truth labels, we conducted experiments using YOLOv8x and YOLOv11x to perform object extraction on COCO images. We perform detection and select objects with a detection confidence score > 0.7 to form a list of ground-truth objects.
>
> Using our framework to fine-tune InstructBlip with the YOLO-based ground-truth objects and evaluating using COCO ground-truth objects, we obtained the following results:
>
> | Model    | Detection mAP | Classification ACC1 | CHAIRi | CHAIRs |
> |----------|---------------|---------------------|--------|--------|
> | Baseline | -             | -                   | 25.8   | 59.1   |
> | YOLOV8   | 53.9          | 79.0                | 7.8    | 25.5   |
> | YOLOV11  | 54.7          | 79.5                | 7.8    | 22.3   |
> | GT       | -             | -                   | 6.8    | 17.8   |
>
> As observed, the hallucination rate is not as low as when using ground-truth objects. However, the results demonstrate that even with less accurate object detectors, our framework effectively reduces hallucination. This also show that the more accurate the object detector is, the better the hallucination reduction that can be achieved.
>
> > 4. More recent baselines should be compared. The baselines (LLAVA, mPLUG-Owl) in Table 4 are from 2023. New versions of LLAVA and mPLUG-Owl are introduced in 2024.
>
> We provide additional results for the latest version of LLaVA (LLaVA-Next):
> | Model   | CHAIRi | CHAIRs |
> |---------|--------|--------|
> | LLaVA-NeXT| 13.0 | 32.0  |
> | GT       | 6.8    | 17.8   |
>
> For the latest mPLUG-Owl3, we encountered bugs during the prediction of the given model, so the results are currently unavailable.
>
> **Question**
> > 1. What is the reason for adopting a different metric for the Genome dataset? why not use F1 or CHAIR?
>
> The CHAIR metric has a limitation due to its reliance on string matching for a fixed set of objects. While this makes it suitable for COCO, it is less accurate for datasets like Visual Genome, which contain a larger number of objects per image and a broader range of object categories. Additionally, we follow the evaluation methodology of HallE-Switch to provide a comparable evaluation that can be referenced in both papers.

---

### Meta-Review · Area_Chair_cwJB · 2024-12-18

**Metareview:**

(a) The paper addresses object hallucinations in LVLMs for image captioning using a human-free RL framework with F1-based rewards and KL divergence constraints. Results show reduced hallucination rates and maintained caption quality.

(b) Strengths:
1) Novel human-free RL approach.
2) Significant hallucination reduction on InstructBLIP and LLaVA (up to 41%).

(c) Weaknesses:
1) Limited evaluation on tasks beyond image captioning.
2) Insufficient comparison with newer baselines (LLaVA-Next).
3) Potential overclaim in title scope.

(d) Decision: Reject. The paper demonstrates promise but lacks comprehensive task generalization and updated baseline comparisons. Title overclaiming undermines focus.

**Additional Comments On Reviewer Discussion:**

Reviewers raised concerns about the paper's scope, lack of task generalization beyond captioning, and outdated baselines. The authors provided new experiments (e.g., LLaVA), addressed metric limitations, and clarified design choices. However, baseline comparisons and broader applicability remained incomplete. These limitations led to reject.

---

### Decision · Program_Chairs · 2025-01-22

Reject